# Genetic and epigenetic intratumor heterogeneity impacts prognosis of lung adenocarcinoma

Xing Hua[1,9], Wei Zhao [2,9], Angela C. Pesatori [3,4], Dario Consonni[4], Neil E. Caporaso[5], Tongwu Zhang [2], Bin Zhu[1], Mingyi Wang [6], Kristine Jones[6], Belynda Hicks [6], Lei Song[6], Joshua Sampson[1], David C. Wedge [7,8], Jianxin Shi[1] & Maria Teresa Landi [2✉]

Intratumor heterogeneity (ITH) of genomic alterations may impact prognosis of lung adenocarcinoma (LUAD). Here, we investigate ITH of somatic copy number alterations (SCNAs), DNA methylation, and point mutations in lung cancer driver genes in 292 tumor samples from 84 patients with LUAD. LUAD samples show substantial SCNA and methylation ITH, and clonal architecture analyses present congruent evolutionary trajectories for SCNAs and DNA methylation aberrations. Methylation ITH mapping to gene promoter areas or tumor suppressor genes is low. Moreover, ITH composed of genetic and epigenetic mechanisms altering the same cancer driver genes is shown in several tumors. To quantify ITH for valid statistical association analyses, we develop an average pairwise ITH index (APITH), which does not depend on the number of samples per tumor. Both APITH indexes for SCNAs and methylation aberrations show significant associations with poor prognosis. This study further establishes the important clinical implications of genetic and epigenetic ITH in LUAD.

[1] Biostatistics Branch, Division of Cancer Epidemiology and Genetics, National Cancer Institute, NIH, DHHS, Bethesda, MD, USA. [2] Integrative Epidemiology Branch, Division of Cancer Epidemiology and Genetics, National Cancer Institute, NIH, DHHS, Bethesda, MD, USA. [3] Department of Clinical Sciences and Community Health, University of Milan, Milan, Italy. [4] Fondazione IRCCS Ca' Granda—Ospedale Maggiore Policlinico, Occupational Health Unit, Milan, Italy. [5] Occupational and Environmental Epidemiology Branch, Division of Cancer Epidemiology and Genetics, National Cancer Institute, NIH, DHHS, Bethesda, MD, USA. [6] Cancer Genome Research Laboratory, Leidos Biomedical Research Inc., Bethesda, MD, USA. [7] Big Data Institute, Old Road Campus, Oxford, UK. [8] Oxford NIHR Biomedical Research Centre, Oxford, UK. [9] These authors contributed equally: Xing Hua, Wei Zhao. ✉email: landim@mail.nih.gov

Lung cancer is the leading cause of cancer mortality, causing more than one million deaths worldwide annually[1]. Lung adenocarcinoma (LUAD) is the most common histologic subtype and accounts for about 40% of lung cancer incidence. While hundreds of LUAD tumors have been profiled extensively based on a single biopsy per patient[2,3], fewer subjects have been investigated for diversity within the tumor through multi-sampling. A few studies have analyzed the extent of intratumor heterogeneity (ITH) of somatic nucleotide variants (SNVs) and/or somatic copy number aberrations (SCNAs)[4–6], and others of DNA methylation[7,8] in LUAD. Some of these studies found a positive association between SNV ITH and risk of relapse[4] or SCNA ITH and cancer free survival (combining risk of mortality and of recurrences)[6]. However, ITH in these studies was estimated without adjusting for the number of samples examined for each tumor and the methylation analysis did not take into account sample purity.

Using multiple samples per tumor it is possible to reconstruct the cancer evolutionary history. In prostate and brain tumors, congruent evolutionary trajectories of genetic and epigenetic mechanisms have been reported[9,10], but to what extent epigenetic changes occur alongside phylogenetic changes in LUAD remains largely unknown.

Here, we perform a comprehensive analysis of ITH of somatic mutations in cancer driver genes, copy number aberrations and DNA methylation in 292 tumor samples from 84 patients with LUAD. We also investigated genetic/epigenetic ITH affecting cancer driver pathways. Moreover, we order genetic and epigenetic events along the LUAD evolutionary trajectories, and test the hypothesis that co-occurrence of genetic and epigenetic mechanisms characterizes the evolution of LUADs. Finally, we assess the clonality of targetable cancer driver genes and evaluate the association of ITH with clinical outcomes (survival and, separately, risk of metastasis) correcting for sample purity and using an unbiased statistical model which takes into account the number of samples examined from each tumor.

## Results

**Patient characteristics.** To determine clonal evolutionary patterns in the genome and epigenome, we performed multi-region sampling from 84 patients with LUAD, of whom 76 (90%) reported past or current smoking. All the samples were treatment-naïve and surgically excised. The demographic characteristics are summarized in Table 1 and Supplementary Data 1. The clinical outcome analyses were based on a median follow-up time of 40.0 months. In total, 292 tumor tissue samples and 157 non-tumor samples (including 74 normal tissue samples, 81 blood samples and 2 buccal cell sample) were collected from 84 subjects. ITH was estimated for tumors that included between 2 and 11 tumor samples for each assay type. For reference, we used blood or buccal cells for deep target sequencing and, to factor out high tissue specificity, normal tissue samples for methylation. For SNP arrays we used only tumor samples. Samples used for each assay type are shown in Fig. 1 and Supplementary Fig. 1.

**Clonal structure of SNVs and SCNAs.** Across all patients, deep target sequencing revealed SNVs in 35 out of the 37 cancer driver genes assayed (Supplementary Data 2). On average, 3.4 genes had nonsynonymous SNVs in each patient. The five most frequently mutated genes were *TP53* (50%), *KRAS* (46.4%), *KMT2C* (39.3%), *STK11* (28.6%) and *KEAP1* (25%), consistent with previous studies for LUAD[2,11,12] (Fig. 2, Supplementary Data 2). For each patient, SNVs were classified as public if all tumor samples from the same tumor carried the SNVs and private otherwise. 65.3%

**Table 1 Distribution of demographic and clinical variables of 84 patients with lung adenocarcinoma.**

| | |
|---|---|
| Age at first diagnosis (mean, range) | 66.3 (49–80) |
| Sex | |
| Male | 68 |
| Female | 16 |
| Smoking status | |
| Never | 7 |
| Former | 34 |
| Current | 42 |
| Cigarettes per day (mean ± s.d.) | 19.1 ± 11.9 |
| ≤10 | 21 |
| >10, ≤20 | 38 |
| >20, ≤30 | 13 |
| >30 | 9 |
| Cigarette smoking duration (mean ± s.d.) | 42.8 + 9.8 |
| ≤30 years | 7 |
| >30, ≤40 | 24 |
| >40, ≤50 | 29 |
| >50 years | 14 |
| Tumor stage | |
| IA | 19 |
| IB | 18 |
| IIA | 13 |
| IIB | 8 |
| IIIA | 21 |
| IIIB | 3 |
| IV | 2 |
| Chemotherapy | |
| Yes | 0 |
| No | 84 |
| Distant metastasis | 44 |
| Deceased | 50 |

(126 of 193) of SNVs were public, a higher proportion than what was observed in the TRACERx study[6] (public SNVs = 50.5%), if we apply the same definition of public vs. private events. In total, 24.3% (47/193; 30 public, 17 private) of SNVs were predicted to strongly alter protein functions (e.g., frameshift or gain of stop codon mutations). The public SNVs showed slightly higher dN/dS ratio than the private SNVs (public SNVs: 3.40 (95% CI: 1.80–6.44), private SNVs: 2.55 (95% CI: 1.29–5.05)). Overall, the functional impact and selective pressure showed no significant difference between public and private SNVs.

Next, we performed unsupervised clustering based on the global SCNA profiles (see Methods section). Intratumoral heterogeneity was lower than intertumoral heterogeneity, with 226/268 (84.3%) samples from the same tumors clustered together and another 7/268 (2.6%) samples from the same tumors in close proximity to each other (Supplementary Fig. 2).

In order to quantify levels of intratumoral heterogeneity, we developed an unbiased metric, the average pairwise ITH index (APITH, see Methods section) and applied it to the SCNA profiles of all patients (Fig. 3a). A major advantage of APITH is that its value is not biased by the number of multi-region samples per tumor while a previously used method[6] is strongly affected (Fig. 3b). APITH ranged from 0 to 0.68 with a mean = 0.184, median = 0.157, and standard deviation = 0.153 (Fig. 3c), suggesting ~18.4% of the genome had different copy number status on average for any pair of tumor samples from the same patient. Of note, the largest APITH values (>0.5) were from patients with only two tumor samples, likely because of large variance in the APITH estimate.

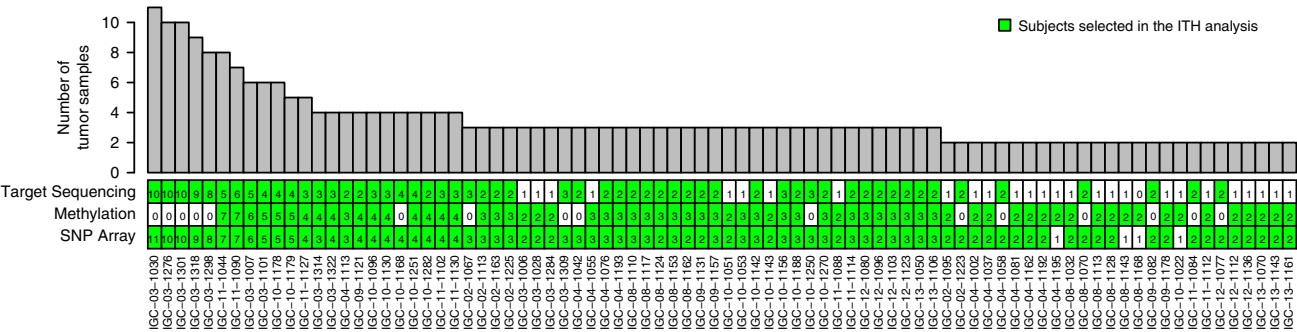

**Fig. 1 Summary of subjects and tumor samples.** Summary of tumor samples that were analyzed on different platforms: ultra-deep targeted sequencing of cancer driver genes (180 tumor samples from 56 subjects), genome-wide methylation (205 tumor samples from 68 subjects) or SNP array profiling (268 samples from 80 subjects). Top panel (bar plot): the total number of tumor samples from each subject. Bottom panel (heatmap): the number of tumor samples from each subject profiled by different platforms. The green color indicates samples assayed by the specific platform were selected for ITH analysis.

Previous TCGA studies[2,13] have reported recurrent SCNA regions for LUAD and identified in these regions 32 candidate driver genes, including 25 amplified and 7 deleted genes. In our samples, SCNA ITH in the recurrently altered regions was similar to SCNA ITH across the whole genome (Supplementary Fig. 3). SCNAs of candidate driver genes were observed in 11.3–72.5% tumors, comparable to the TCGA study (Supplementary Data 3). *SMARCA4* deletion and *MCL1* amplification were the most frequent SCNA events (72.5% and 68.75%, respectively). *KRAS* and *EGFR* amplification were also observed in 38.75% and 41.25% of tumors, respectively, of which 41.9% and 33.3% were public.

We do not report APITH of SNVs since we conducted target sequencing of 37 cancer driver genes only, with resulting low number of mutations identified.

**Intratumoral heterogeneity of DNA methylation**. We first performed unsupervised hierarchical clustering based on methylation profiles using the 5000 most variable CpG sites across the genome (Fig. 4a) and, separately, limited our analysis to CpGs in promoter regions (Supplementary Fig. 4). Both analyses confirmed that normal tissue samples from almost all subjects (59/61) clustered together. Similarly, 183/205 (89.3%) samples from the same tumors clustered together, showing higher intertumoral heterogeneity than intratumoral heterogeneity.

Previous studies have identified high levels of methylation in promoter regions of some genes, also referred to as the CpG island methylator phenotype (CIMP) in multiple cancer types, including lung cancer[2,14,15]. 16/68 (23.5%) patients had significantly altered CpG island methylator phenotype (CIMP-H) and 47/68 (69.1%) had a normal-like pattern (CIMP-L). In five patients, both CIMP-H and CIMP-L patterns were observed in the same tumor.

We next examined the distribution of DNA methylation ITH based on either the probes across the whole genome or those mapping to specific genomic regions (Fig. 4b). The CpG probes mapping to CpG island regions had a significantly lower APITH compared to those mapping to other regions ($p = 1.09 \times 10^{-10}$), as previously observed in aggressive prostate cancer[10]. Moreover, the CpG probes mapping to gene promoter regions (TSS1500, TSS200, 5′ UTR and first exon) had lower APITH compared to those mapping to gene bodies, 3′ UTR regions and intergenic regions (t-test $p = 1.626 \times 10^{-8}$).

Restricting the analysis to CpG probes mapping to 250 oncogenes and 300 tumor suppressor genes predicted by TUSON Explorer[16] revealed that methylation ITH mapping to tumor

suppressor genes was significantly lower than that of oncogenes (t-test $p = 1.68 \times 10^{-17}$) and that of other genes (t-test $p = 1.50 \times 10^{-16}$) (Fig. 4c). Inactivation of tumor-suppressor genes by hypermethylation at promoter regions has been observed in multiple cancer types including lung cancer[17–19]. Lower DNA methylation ITH in these regions suggests greater selective pressure which is consistent with their high putative impacts in oncogenic transformation.

**ITH of genomic alteration types in cancer driver pathways**. We analyzed genetic and epigenetic aberrations of 13 cancer driver genes in the RTK/RAS/RAF pathway that are frequently mutated in LUAD[2] (Fig. 5). For SCNAs, we only included amplification of oncogenes and deletion of tumor suppressor genes. For DNA methylation, we determined abnormality based on probes located in CpG islands at promoter regions of the target genes.

Across the 13 genes, 77/84 (91.7%) tumors harbored genetic or epigenetic alterations in this pathway; ITH was observed in 69 (89.6%) tumors. SNVs, SCNAs, and methylation of the driver genes altered 7.28%, 17.5%, and 4.19% of the tumors, respectively. Different types of genetic or epigenetic alterations affected different samples in the same tumor. For example, in tumor IGC-11-1130, four samples were tested and all had alterations in the *KRAS* pathway. Among them, two samples had amplification in *ROS1* and the other two samples had aberrant DNA methylation in the promoter regions of *ALK* (Fig. 5).

In the cell cycle pathway, genetic or epigenetic changes in *RB1*, *CDKN2A*, or SWI/SNF components were observed in 57/84 (67.9%) tumors, with ITH in 33 (57.9%) tumors.

**Evolutionary trajectories of genetic and epigenetic events**. To reconstruct the evolutionary trajectories, we inferred clonal relationships for tumors that were assayed for both SCNA and DNA methylation in multiple regions. For each pair of tumor samples per tumor, we first calculated the Euclidean distance separately for SCNA profiles and the DNA methylation levels of the 5000 most variable CpG probes. We found that the pairwise SCNA distances were positively correlated with DNA methylation distances (Fig. 6a, Spearman's correlation coefficient's = 0.586, $p < 1 \times 10^{-16}$). To exclude the possibility that the DNA methylation changed purely as a consequence of the changed ratio between alleles, we carried out a sensitivity analysis by testing the correlation between SCNA and methylation distances only in regions with the same copy number status (gain/loss/neutral) for all samples from the same tumor, defined as SCNA homogeneous regions, and also in copy number neutral regions. Both analyses

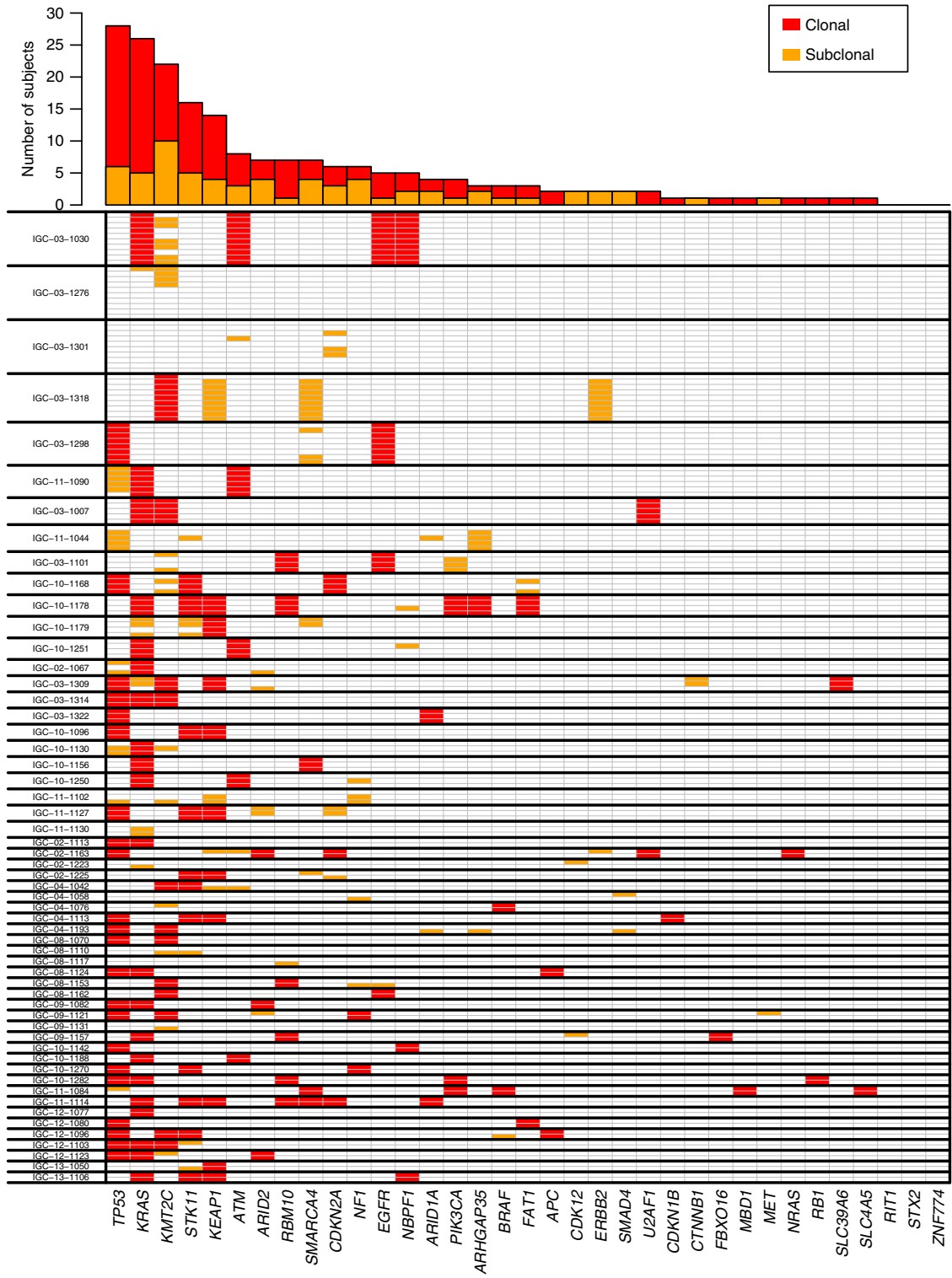

**Fig. 2 Intratumor heterogeneity of single nucleotide variants for 35 lung driver genes.** Only nonsynonymous mutations are included for analysis. Top panel: the number of public (shared by all tumor samples in a patient) and private (not shared by all tumor samples in a patient) mutations for each driver gene. Bottom panel: summary of intratumor heterogeneity for each gene in each tumor. Thick lines separate the different tumors. Multiple lanes within thick lines indicate multiple samples from the same tumor. Red and orange indicate public and private mutations, respectively.

showed consistent results and supported the congruence between SCNAs and methylation (Supplementary Fig. 5, panels A–D for 14 tumors with at least four samples each assayed for methylation, Spearman's correlation coefficient's = 0.638 and 0.573 for SCNA homogenous and copy-number neutral regions, respectively, $p < 1 \times 10^{-16}$). We then examined the topology of

evolutionary trees inferred from SCNAs and DNA methylation and observed high similarity (Fig. 6b for six tumors with at least five samples each; Supplementary Fig. 5E for the same analysis based on CpG probes across the whole genome, CpG probes mapping to SCNA homogenous regions, and CpG probes mapping to copy number neutral regions; and Supplementary Fig. 6

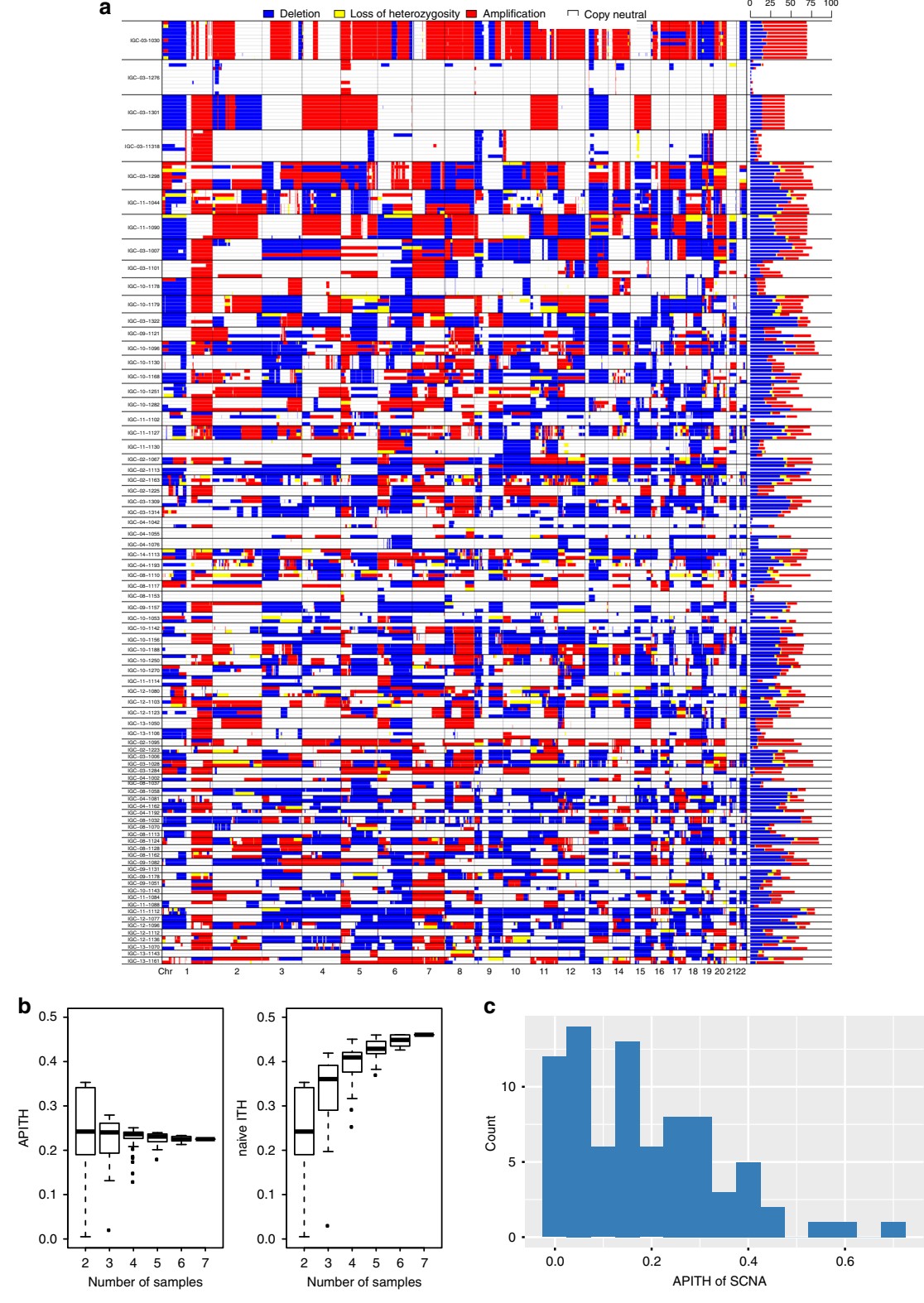

**b**

APITH vs Number of samples; naive ITH vs Number of samples

**c**

Histogram: Count vs APITH of SCNA

for eight tumors with at least four samples each). These results support the co-occurrence of the two mechanisms in shaping the cancer genome.

Of note, when we overlaid the SNVs in target genes on the SCNA-derived and methylation-derived trees, we observed that some genes were altered by different mechanisms in the same trees. For example, tumor IGC-10-1179 had *STK11* mutated in the trunk and deleted in a branch (Fig. 6c).

**Associations between ITH and clinical outcomes**. We tested the association of SCNA and DNA methylation APITH with clinical data and observed no significant correlations with age, tumor stage, or grade (Supplementary Data 4). Smokers had higher APITH of SCNAs (*t*-test $p = 0.035$, nominally significant) but similar APITH of methylation compared with non-smokers. Other smoking behaviors (e.g., smoking intensity and duration) were not associated with APITH of SCNA or DNA methylation.

**Fig. 3 Intratumor heterogeneity of somatic copy number alterations (SCNAs). a** Left panel: SCNA landscape of 80 subjects with multi-region sampling. Right panel: fractions of genomes disrupted by a specific type of SCNA. Colors represent different SCNA event types: blue indicates deletion, yellow indicates loss of heterozygosity, red indicates amplification, and white indicates copy number neutral. **b** APITH score (left) and naïve (right) calculated based on the tumor samples of subject IGC-11-1044 with seven tumor samples. For a given set of tumor samples, APITH was calculated as the average pairwise distance between any pair of tumor samples; naïve ITH index was calculated as the fraction of genome disrupted by private SCNAs that were not shared by all tumor samples. For a given number ($K = 2, ..., 7$) of tumor samples, we numerated all combinations of $K$ tumor samples to derive the distribution of ITH index. The naïve ITH index positively depends on the number of tumor samples while APITH does not. The center line in the box plots indicates median APITH or naïve ITH index. The box length indicates the interquartile range (IQR). The whiskers extend to the largest and smallest APITH or naïve ITH at most 1.5*IQR. **c** Distribution of pairwise average ITH of SCNAs for 80 subjects with average APITH score 0.184.

Next, we examined the associations between APITH and survival or risk of distant metastasis. For each analysis, we performed a Cox proportional-hazards model weighted or unweighted by the variance of the estimated APITH. For significant associations, we found that weighted analysis had smaller $p$-values than unweighted analysis, as expected. Thus, we report below results based on weighted analyses (summary of weighted and unweighted results is in Supplementary Datas 5 and 6).

Similar to the TRACERx study[6], we found that increased SCNA APITH was associated with poor overall survival with $p = 0.05$ using all patients and $p = 0.0044$ (HR = 1.77, 95% CI = 1.2–2.6) when restricting the analysis to patients with ≥ 3 tumor samples per tumor (Supplementary Data 5 and Fig. 7a). SCNA APITH was not significantly associated with risk of developing distant metastases (Supplementary Data 5). Of note, for both overall survival and distant metastasis, APITH based on SCNAs in the 37 cancer driver genes provided lower prognostic value than the APITH of SCNAs in the whole genome.

We then tested the association of DNA methylation-based APITH with overall survival and the risk of distant metastases. We found that APITH based on the 5000 most variable CpG probes was associated with overall survival (HR = 1.27, 95% CI = 1.05–1.55, $p = 0.016$, Supplementary Data 6 and Fig. 7b) but not significantly associated with risk of metastasis ($p = 0.14$). APITH based on CpG probes mapping to island regions had the strongest association with overall survival (HR = 1.31, 95% CI = 1.10–1.57, $p = 0.0028$, Fig. 7c) and were also found to be associated with risk of distant metastasis (HR = 1.35, 95% CI = 1.07–1.72, $p = 0.012$, Fig. 7d). The results for APITH defined based on other genomic regions are in Supplementary Data 6. The CIMP phenotype did not show substantial ITH, i.e., there were only a few tumors with CIMP-H and CIMP-L across samples from the same tumor. Therefore, we could not analyze the association of APITH of CIMP with clinical outcomes.

## Discussion
In this study, we investigated genetic and epigenetic intra-tumor heterogeneity based on multi-region sampling per tumor across 84 patients with LUAD. On average, 35% of SNVs in targeted genes were private and ~18.4% of the genome had SCNA ITH for any pair of samples from the same tumor. Methylation in CpG islands or gene promoter regions, particularly of tumor suppressor genes, had low ITH. Different types of somatic alterations across samples from the same tumors affected cancer driver genes in the RTK/RAS/RAF or cell cycle pathways. SCNAs and DNA methylation changes showed congruent evolutionary trajectories. Notably, we developed a statistical approach to correctly estimate ITH for any pair of tumor samples from the same patient and showed that ITH of SCNAs and DNA methylation was associated with poor prognosis.

The findings of substantial ITH across different genomic types and of similar tumor evolutionary trajectories for genetic and epigenetic changes are important to understand the biology and natural history of LUAD. Moreover, they are crucial to inform clinical management and therapeutic strategies. Using multi-region sampling, we identified private events in cancer driver genes, which may not be detected by a single biopsy or by limiting the analyses to point mutations. For example, combining genetic and epigenetic changes, 95.2% of tumors had activating events in the RTK pathway, of which ~36% were private.

ITH of SCNA and ITH of DNA methylation (overall and in CpG islands) were similarly associated with shorter survival in our study, and ITH of methylation in CpG islands was also associated with higher risk of developing metastasis. Adding both measures in the same model did not significantly improve the prediction value (data not shown), likely because the two measures were highly correlated to each other.

In previous studies, ITH of SCNAs was quantified as the fraction of SCNAs not shared by all samples in the tumor[6]. Clearly, this ITH index positively depends on the number of tumor samples per tumor (Fig. 3b) and thus hinders valid cross-patient comparisons or testing associations with clinical outcomes. Moreover, unobserved factors that are associated with the number of tumor samples per patient, e.g., tumor size or different study sites, may confound the association analysis with clinical outcomes. We propose APITH as an index for ITH, similar in spirit to that previously proposed for quantifying ITH from the observation of SNVs[20]. This index, defined by pairwise distances of genomic profiles, is not biased by the number of samples per tumor and thus allows association testing for any genomic profiling platforms. Crucially, the variance of APITH estimates depend on the number of samples per tumor and thus a naïve statistical association test between APITH and any outcome may have low statistical power. We explicitly addressed this issue by quantifying the variance of APITH and proposing a procedure for its numerical calculation. This variance was used to weight subjects in the regression analyses to achieve the best statistical power, as demonstrated in theoretical analyses (Supplementary Methods). As a confirmation, our empirical results showed that the weighted analyses produced more significant results than the unweighted analyses for the association between APITH and overall survival.

A comparison between the previous TRACERx study[6] and this study using APITH to estimate the ITH of SCNAs is reported in the Supplementary Notes.

In conclusion, our results delineate the genetic and epigenetic ITH in LUAD and provide a rigorous statistical approach to estimate ITH for comparisons across individuals and for associations with clinical outcomes. DNA methylation and genomic changes followed similar evolutionary trajectories and strongly impacted cancer driver genes and pathways in a complex manner. These findings can inform the clinical management of LUADs suggesting that taking multiple biopsies and analyzing multiple genomic types may be needed to capture the landscape of targetable events. Future larger studies are warranted to identify the combination of genomic types and genomic regions to best predict clinical outcomes.

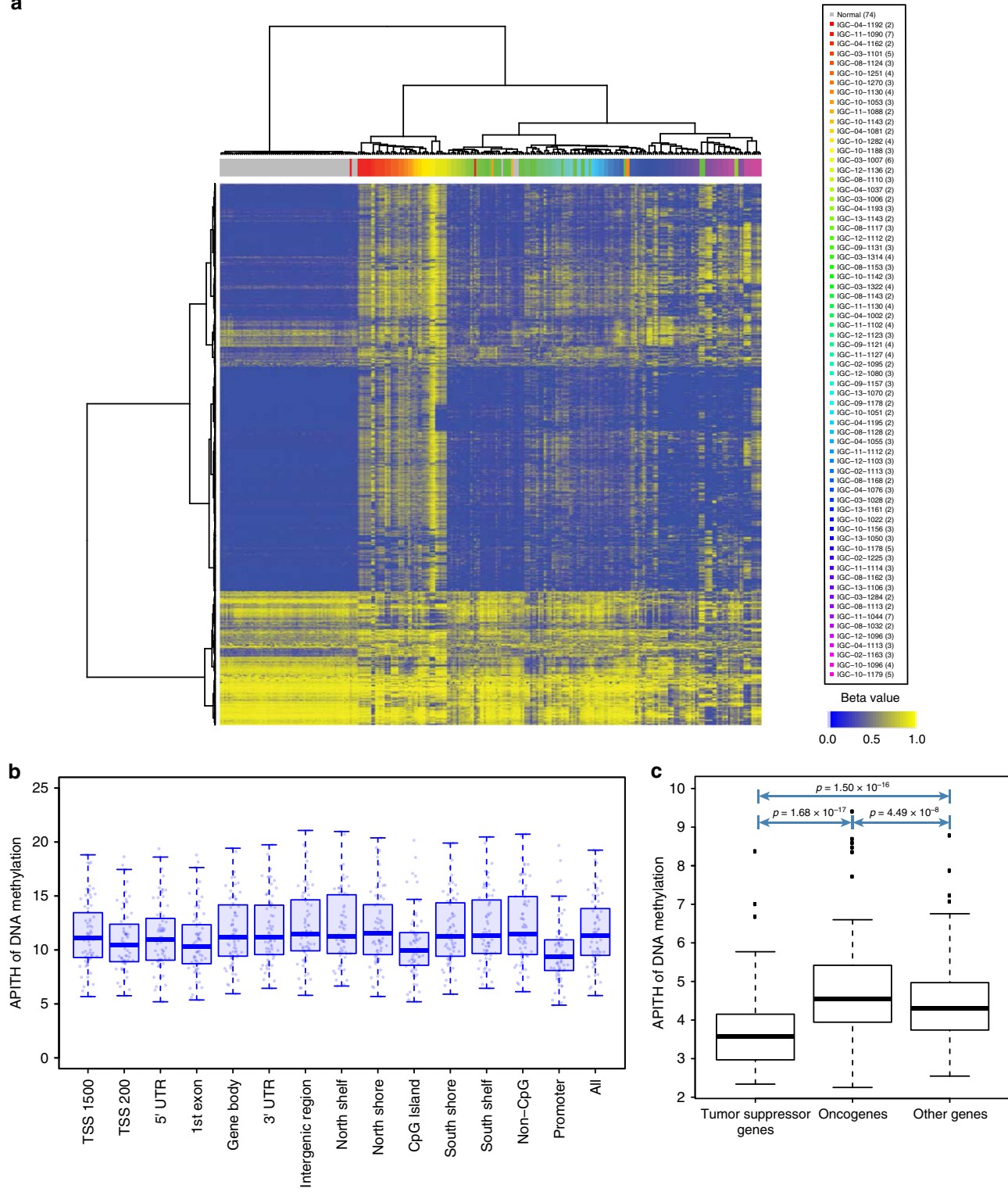

**Fig. 4 Intratumoral heterogeneity of DNA methylation profiles. a** Unsupervised hierarchical clustering of 5000 most variable probes in CpG islands of the genome in 68 subjects. Different tumors are indicated by different colors in the column sidebar, with normal samples colored in gray. The numbers in parenthesis are the number of normal tissue samples for the 'normal' group, or the number of tumor samples in each patient. The beta values represent estimates of methylation levels, with 0 being unmethylated and 1 fully methylated. **b** Distribution of ITH of DNA methylation in different genomic contexts. TSS 1500: 200–1500 bases upstream of the transcription start sites (TSS), TSS 200: 0–200 bases upstream of the TSS. 5'UTR: Within the 5' untranslated region, between the TSS and the ATG start site. Gene body: Between the ATG and stop codon. 3'UTR: From the stop codon to poly A tail. Island: CpG island. Shore: 0–2 kb from island. Shelf: 2–4 kb from island. North: upstream (5') of island. South: downstream (3') of island. **c** ITH of DNA methylation in oncogenes ($n = 176$), tumor suppressor genes ($n = 223$) and other genes ($n = 12,837$). The p-values are based on unpaired two-sided t-test of the two groups indicated by arrows. The center line in the box plots indicates median APITH. The box length indicates the interquartile range (IQR). The whiskers extend to the largest and smallest APITH at most 1.5*IQR.

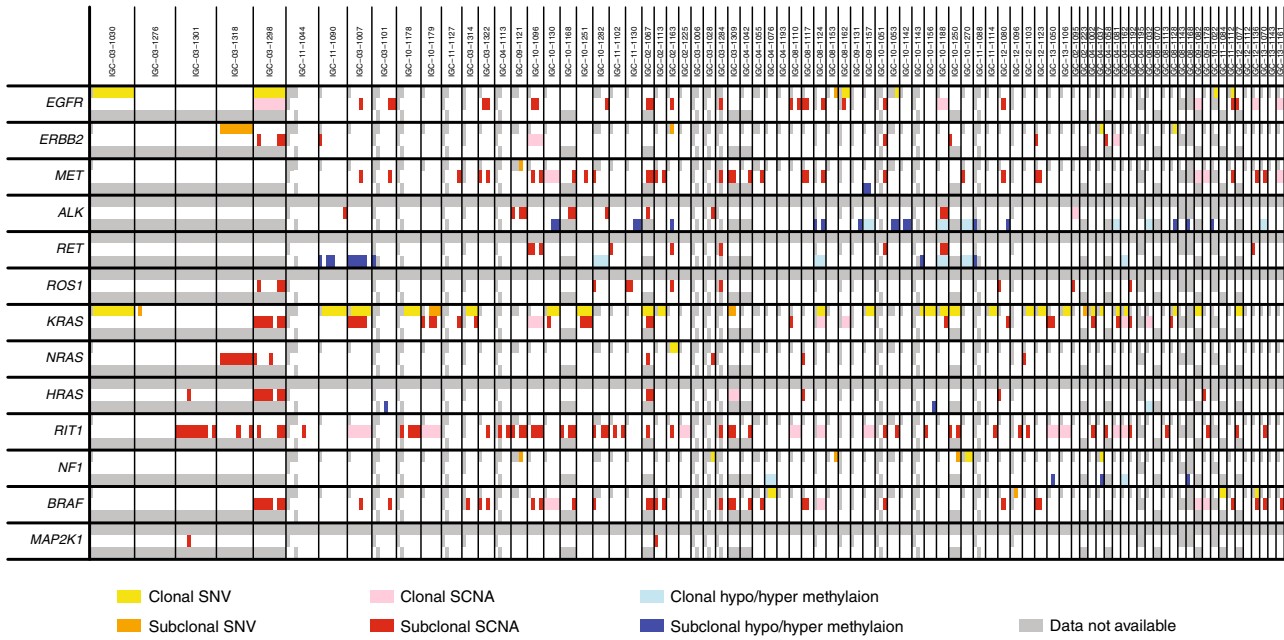

**Fig. 5 Intratumor heterogeneity of genomic and epigenomic alterations of 13 cancer driver genes in RTK/RAS/RAF pathway.** Shown are public and private SNVs, SCNAs, and DNA methylation alterations in 84 subjects. SNVs, SCNAs, and DNA methylation alterations are indicated by yellow/orange, red, and blue, respectively. Clonal and subclonal events are indicated by light and dark shades, respectively.

## Methods

**Patients and multi-region tumor samples.** The current work includes 84 patients with LUAD from the EAGLE study, a population-based case-control study conducted in Italy between 2002 and 2005 (refs. [12,21,22]). Samples were snap-frozen in liquid nitrogen within 20 min of surgical resection and the precise site of tissue sampling was recorded. All tumor samples were histologically confirmed as primary LUAD (not mixed types or undifferentiated cases) with at least 50% tumor nuclei and <20% necrosis. Normal tissue samples were taken at > 3 cm from the tumor tissue and had to have no tumor nuclei at histological examination. Based on these characteristics and DNA quality, we were able to analyze between 2 and 11 samples from each tumor, for an overall 292 tumor samples and 157 non-tumor samples (including 74 normal tissue samples, 81 blood samples and 2 buccal cell samples). Detailed information on tumor characteristics, recurrence, treatment, and follow-up data were recovered from patients' medical records and follow-up visits and hospital admissions were identified by linkage with the region-wide Regional Health Authority database (Supplementary Data 1). Recurrence history was ascertained through December 31, 2010. The study protocol was approved by the Institutional Review Board of the US National Cancer Institute and the involved institutions in Italy, including A.O. Ospedale Niguarda Cà Granda—Milano, A.O. Spedali Civili—Brescia, Istituto Clinico Humanitas—Rozzano (Milano), Ospedale di Circolo e Fondazione Macchi—Varese, Fondazione IRCCS Ospedale Maggiore Policlinico, Mangiagalli e Regina Elena—Milano, Istituto Scientifico Universitario Ospedale San Raffaele—Milano, A.O. Ospedale Luigi Sacco—Milano, A.O. San Paolo—Milano, A.O. Ospedale San Carlo Borromeo—Milano, IRCCS Policlinico San Matteo—Pavia, A.O. San Gerardo—Monza, A.O. Ospedale Fatebenefratelli e Oftalmico—Milano, and Ospedale San Giuseppe—Milano. Informed consent was obtained for all subjects prior to study participation.

**Genomic and epigenomic profiling of tumor samples.** Somatic copy number alterations (SCNAs) in tumors were profiled using 705,667 probes from the Illumina HumanOmniExpress SNP arrays in 268 tumor samples from 80 subjects. SCNA segmentation was performed using ASCAT[23] based only on B allele frequencies to avoid hypersegmentation; copy number status of each segment was determined based on log R ratio. Bisulfite treatment and Illumina Infinium HumanMethylation450 BeadChip assays were performed to derive the DNA methylation levels in 205 tumor samples and 74 normal tissue samples from 68 subjects; in the absence of paired normal tissue samples for seven tumors, methylation levels were imputed based on the methylation levels of the overall normal samples. For DNA methylation, we analyzed 338,730 CpG probes after excluding probes annotated with genetic variants, in repetitive genomic regions or on the X-chromosome. Deep target sequencing was performed to identify SNVs for 37 established lung cancer driver genes[2,11,12] with an average sequencing depth of 500X in 180 tumor samples and 55 blood samples and 1 buccal sample from 56 subjects. Sequence data were processed using the standard Ion Torrent Suite Software (Thermo Fisher Scientific) version 5.0.7 and somatic mutations were detected using Torrent variant caller (TVC) version 5.0.9. More details can be

found in Supplementary Methods. Samples used for each assay are listed in Fig. 1 and Supplementary Fig. 1.

**Bioinformatic analyses of DNA methylation data.** Bisulfite treatment and Illumina Infinium HumanMethylation450 BeadChip assays were performed to derive the DNA methylation levels. Raw methylated and unmethylated intensities were background-corrected, and dye-bias-equalized, to correct for technical variation in signal between arrays. For background correction, we applied a normal–exponential convolution, using the intensity of the Infinium I probes in the channel opposite their design to measure non-specific signal. For each CpG probe, the DNA methylation level was summarized as the fraction of signal intensity obtained from the methylated beads over the total signal intensity. After excluding CpG probes annotated with genetic variants, in repetitive genomic regions or on the X-chromosome, 338,730 CpG probes remained for analysis. Each CpG probe was annotated as in CpG Island (denoted as CGI), nonCGI (including shores and shelves) or open-sea. Each CpG probe was also annotated as in promoter (TSS200, TSS1500, and first exon), body, 3′UTR in a specific gene or annotated as intergenic. Methylation ITH of specific genomic regions was computed using the average pairwise distance of the top 10% variably expressed probes mapping to that region scaled by the number of probes. To identify potential driver DNA methylation events, we analyzed CpG island regions of cancer driver genes and compared the beta values of tumor samples and corresponding normal samples. We used 0.3 as the cutoff value to call differences in beta values[7].

**Bioinformatic analyses of deep target sequencing data.** Deep target sequencing was performed to identify SNVs for 37 established lung cancer driver genes with an average sequencing depth of 500×. The genes were targeted with an Ion Ampliseq panel, and enriched libraries were sequenced using P1 chips on the Ion Proton sequencer. All laboratory analyses were performed at the Cancer Genomics Research Laboratory (CGR) of the Division of Cancer Epidemiology and Genetics, NCI. Sequence data were processed using standard Ion Torrent Suite Software (Thermo Fisher Scientific) version 5.0.7. The data processing pipeline includes signal processing, base calling, quality score assignment, adapter trimming, read alignment to hg19, coverage analysis and somatic variant calling. We used TVC version 5.0.9 to detect somatic mutations. A somatic mutation was detected if the variant allele count >3, coverage >2 in both tumor and normal samples and variant allele fraction ≥ 0.1. The dN/dS ratio was estimated using R package *dNdScv*[24].

**Bioinformatic analyses of SCNA data.** We initially performed SCNA analysis using ASCAT[23] with default parameters, which uses both LRR (log R ratio) and B-allele frequency (BAF). Extensive SCNAs and very complex subclonality patterns made segmentation difficult. Thus, we modified the ASCAT to rely only on BAFs for segmentation. Segments with BAF values different from 0.5 were identified as SCNA regions. We compared LRR values between SCNA regions and segments with BAFs = 0.5 using the Student's t-test. If the SCNA region LRRs were

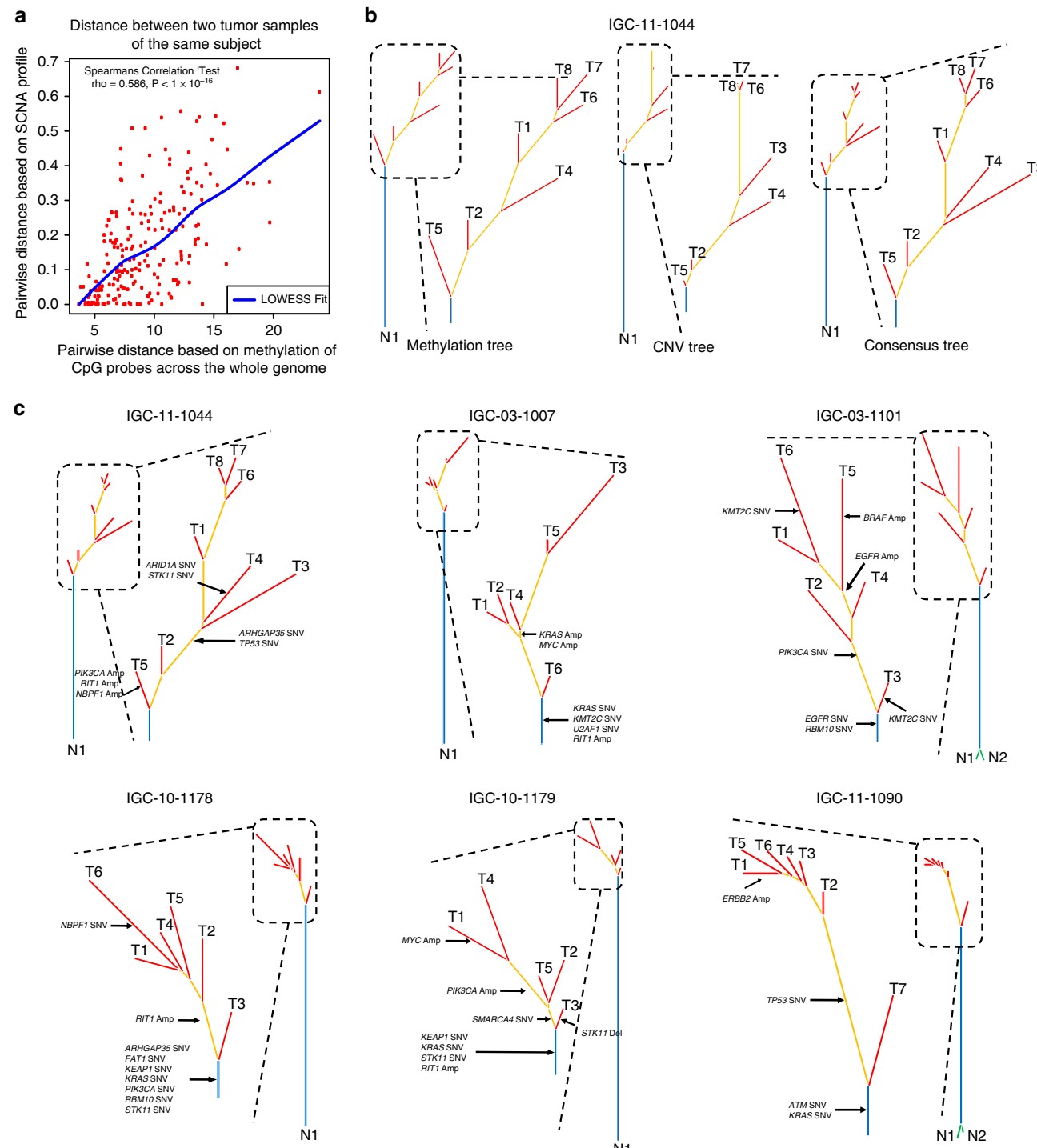

**Fig. 6 Reconstruction of evolutionary trajectories from SCNA and DNA methylation profiles. a** Pairwise distance of tumor samples from the same tumor based on DNA methylation and SCNA profiles. Each dot represents a pair of tumor samples from the same subject. Significant Spearman's correlation coefficient is shown ($n = 212$ tumor sample pairs). **b** Phylogenetic analysis of subject IGC-11-1044 based on DNA methylation and SCNA profiles, and the consensus phylogenetic tree built based on the distance incorporating both SCNA and DNA methylation profiles. Blue lines represent alterations shared by all tumor samples from the same subject. Yellow lines represent alterations shared by two or more tumor samples. Red lines represent alterations specific to one tumor sample. Green lines represent alterations specific to one normal sample. **c** Consensus phylogenetic trees for six tumors with at least five samples assayed for both SCNAs and DNA methylation. SNVs, deletions of tumor suppressor genes and amplifications of oncogenes are marked on the inferred phylogenetic tree.

significantly higher or lower at significance level of 0.05 after adjusting for multiple comparisons, the segments were identified as amplified or deleted, respectively. Otherwise they were identified as LOH. In addition, amplifications with at least four copy numbers in oncogenes and deletions with zero copy number in tumor suppressor genes were identified as potential driver events.

**Statistical analyses**. Regional genetic and epigenetic evolutionary trees for each patient were built using *fastme.bal* in an R package *ape* that uses the minimum evolution algorithm based on a distance matrix of SCNA or methylation profiles[25]. The consensus tree was built using a revised distance matrix combining both SCNA and DNA methylation profiles with methods described briefly below. Let $d_{ij}^1$

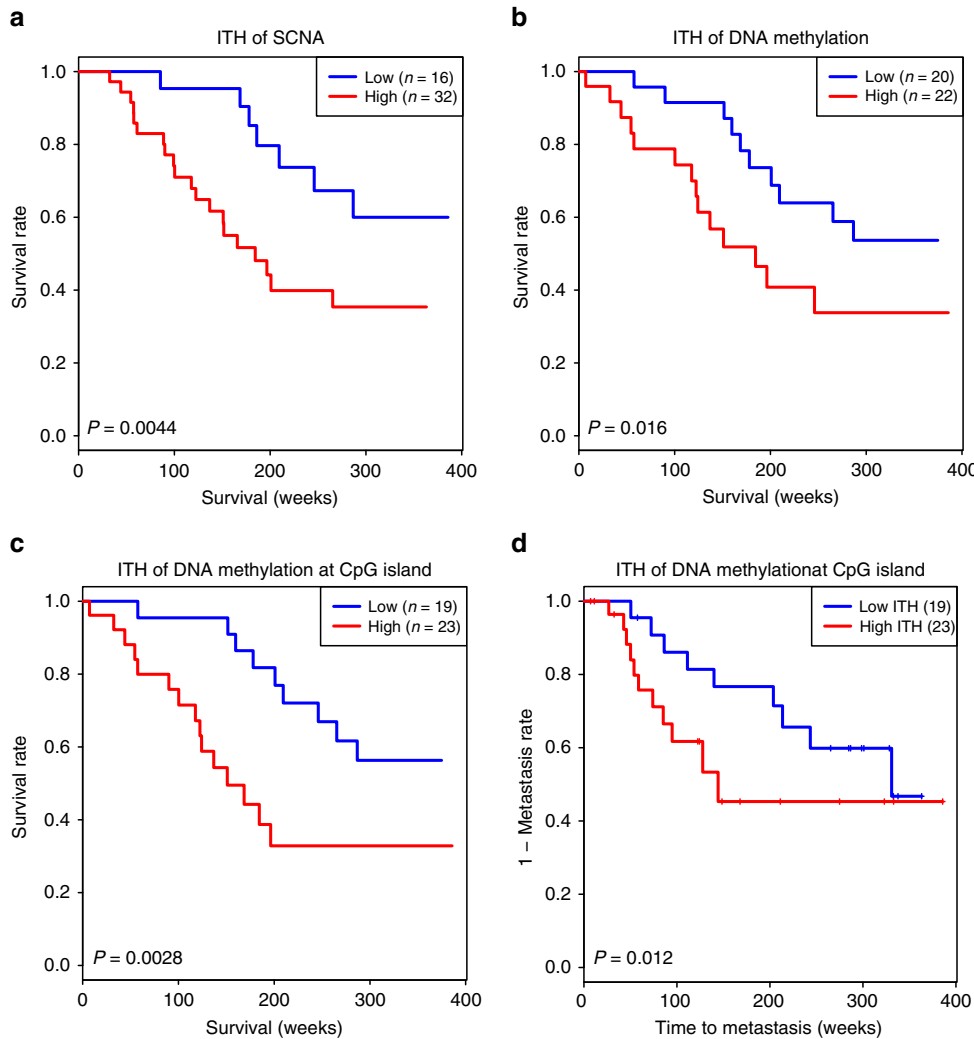

**Fig. 7 Kaplan–Meier curves of overall survival.** Kaplan–Meier estimates of overall survival in patients with high and low ITH of **a** SCNAs, **b** DNA methylation based on the top 5000 most variable CpG probes and **c** DNA methylation at CpG islands. **d** Kaplan–Meier estimates of metastasis in patients with high and low ITH of DNA methylation at CpG islands. High and low ITH groups were stratified by median APITH and colored in red and blue, respectively. The p-values were calculated using the Cox proportional-hazards model weighted by the variance of the estimated APITH. Sample size for each group is indicated in the figure.

denotes SCNA distance and $d_{ij}^2$ denotes methylation distance. When both samples have SCNA and methylation profiles, the new distance was defined as $d_{ij}^0 = 0.5\left(\frac{d_{ij}^1}{\max d_{kl}^1} + \frac{d_{ij}^2}{\max d_{kl}^2}\right)$. When only SCNA profiles are available, we define $d_{ij}^0 = \frac{d_{ij}^1}{\max d_{kl}^1}$. Here, the denominators are used to rescale distances so that they are comparable between SCNA and DNA methylation profiles.

The analysis of the CIMP was performed using the hierarchical clustering based on the 5000 most variable CpG probes mapping to gene promoter regions and CpG island regions.

**Methylation analysis adjusted by sample purity.** For a given CpG probe, the observed DNA methylation was a linear combination of the data from the normal and the tumor tissue samples weighted by tumor purity. ITH may be overestimated if purity varies across tumor samples from the same patient. Thus, we estimated purity, $\pi$, for each tumor sample and derived the purity adjusted methylation values using R package *InfiniumPurify*[26]. All downstream methylation analyses were based on purity-adjusted methylation.

**Quantification of ITH.** The frequently used index[6] that measures ITH for a patient using the fraction of aberrations present in all samples positively depends on the number of multi-region tumor samples. This estimate may vary across tumors making the association analysis between ITH and clinical outcomes problematic. To address this problem, we defined an ITH metric, average pairwise ITH or APITH, for each patient. For a patient with $k$ tumor samples, and with $d_{ij}$ defined as the genomic or epigenetic distance between a pair of samples $(i, j)$, the APITH is defined as the average across all pairs of samples:

$$\text{APITH} = \frac{2}{k(k-1)} \sum_{1 \le i < j \le k} d_{ij}. \tag{1}$$

The expectation of APITH does not depend on the number of multi-region tumor samples. For SCNAs, $d_{ij}$ is calculated as the proportion of the 705,667 probes with different copy number status for $(i, j)$. To investigate ITH of lung cancer driver genes[2,11,12], we also calculated $d_{ij}$ as the fraction of the genome in these driver gene regions with differing copy number state. For DNA methylation, we define $d_{ij}$ as the Euclidean distance calculated for a given set of CpG probes, e.g., all CpG probes after quality control (QC), CpG probes mapping to CpG island and gene promoter regions. These analyses are informative to investigate whether ITH defined based on specific genomic regions would be useful for predicting prognosis.

The variance of the APITH estimate depends on the number of multi-region tumor samples. Intuitively, APITH is more accurate for patients with more multi-region tumor samples and should be weighted up in downstream statistical analyses to optimize statistical power. As described in Supplementary Methods, we heuristically derived sample weights that were used for the weighted Cox proportional-hazards model using *svycoxph* in the R package *survey*[27] to investigate the association between APITH and clinical outcomes (overall survival and the risk of distant metastasis). The survival analysis was adjusted for stage, age, and smoking status. KM-plot were stratified by the median APITH of subjects with at least two tumor samples.

**Reporting summary.** Further information on research design is available in the Nature Research Reporting Summary linked to this article.

## Data availability

The target sequencing data have been deposited in SRA through dbGaP under the accession number phs001169.v2.p1. The SNP array and methylation array data have been deposited in dbGaP under the same accession number. All other data are available tin the Article, Supplementary Information or available from the author upon reasonable request.

## Code availability

The corresponding R code has been distributed at https://github.com/xtmgah/EAGLE_LUAD.

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

## Acknowledgements

This work was supported by the Intramural Program of the Division of Cancer Epidemiology and Genetics, National Cancer Institute, NIH, and utilized the computational resources of the NIH high-performance computational capabilities Biowulf cluster (http://hpc.nih.gov). We are grateful to the patients and families who contributed to this study and the many investigators who are involved in the EAGLE study. We also thank the National Cancer Institute Cancer Genomics Research Laboratory (CGR) for sample preparation and quality control laboratory analyses. D.C.W. is funded by the Li Ka Shing Foundation and the National Institute for Health Research, Oxford Biomedical Research Centre.

## Author contributions

M.T.L. conceived the study. A.C.P., D.C., N.E.C. oversaw the surgical sampling design, sample and data collection, and field activities. X.H. and J.S. performed the statistical analysis of phylogenetic trees, developed the statistic for intra-tumor heterogeneity analyses, and conducted association analyses. W.Z., X.H, and T.Z. conducted the bioinformatics analyses. J.Sa. and B.Z. helped with the statistical analyses. L.S. helped with the bioinformatic analyses. M.W., K.J., and B.H. conducted the laboratory analyses. W.Z., D.C.W., J.S., and M.T.L. discussed the results and implications and wrote the manuscript. All authors reviewed the manuscript.

## Competing interests

The authors declare no competing interests.
