## [Peer Review File · Nature Communications]

Reviewers' comments:

Reviewer #1 (Remarks to the Author):

In this interesting study, Hua et al investigated ITH of LUAD at SNV, SCNA and methylation level. Although the findings are similar to what have been reported by other groups, the development of the APITH approach is novel and could be a useful tool, particularly for studies on multiregion profiling of various number of samples per tumor. As such, this study is potentially interest to the community of heterogeneity of lung cancers. However, there are some technical and conceptual concerns.

1. Overall, the manuscript was well written and easy to follow. However, it may be too "concise" with inadequate information to assess the validity of the analyses. For example, it was not clear whether germ line DNA was used for SCNA analysis. It appears that SNP array was only applied in tumor sample. If so, it may not be appropriate to do SCNA segmentation using ASCAT without normal samples.
2. There are many interesting findings. However, the authors just described the data, but did not interpret the meaning. For example, the authors discovered "revealed that methylation ITH mapping to tumor suppressor genes was significantly lower than that of oncogenes and other genes (line 135-136)". What is the interpretation of this finding?
3. In the unsupervised clustering with SCNA and DNA methylation profiles, how many patients showed the concordant clustering (all the regions of the same tumor cluster together)? It looks majority, but how many? This should be described in the text.
4. The figure legends for supplemental figures are too simple, which made it different to interpret the data.
5. The terms clonal and subclonal were not appropriate here. Since the authors did not do subclonal analysis to derive cancer cell fraction, they can not say whether the changes are clonal or subclonal. Based on the definition, trunk and branch are more appropriate terms.
6. It is not fair to compare the % of clonal vs subclonal SNVs with TRACERx study as the latter used PyClone to derive the clonality of SNVs. The authors can calculate the % trunk vs branch events from TRACERx publications for a fair comparison.
7. Ion Torrent is a good technology to validate known mutations. But it is not the best approach for "discovery" because of its much higher error rate. As such, one has to be very stringent to avoid technical artifacts. The filtering criteria "the variant allele count >3, coverage >2 in both tumor and normal samples and variant allele fraction ≥ 0.1 " was relatively loose, which may lead to concerning proportion of SNVs are false positive mutations. There are some additional options. 1) Authors could use other software to call mutations and only include mutations called by more than one caller. 2) Authors should increase the sequencing depth and alternative read cut offs to increase the reliability. 3) Authors should pull out the TCGA data regarding these 37 genes to calculate the average number of non-synonymous mutations per sample to validate the findings are comparable.
8. What is the definition of gain or loss? Is this based on Log Ratio? What cut off did you use, 0.3, 0.6, 1 or what?
9. The authors adjusted the methylation data by purity. What about SNCA data? Purity impacts SNCA profile too.
10. From deep-targeted sequencing, they found SNVs from 35 cancer driver genes. Are they all "cancer driver mutations"? Although they occurred in cancer driver genes, most of them can be passengers, not a driver mutation. It would be interesting if we can classify the mutations into "drivers" and "passengers" first and check what proportion of them is "clonal" and "subclonal".
11. The authors claimed that the proportion of clonal mutations was lower than in a previous TRACERx study. How did they deal with regions with low purity? Did they have any kind of rescue strategy for the mutations missing in the low purity samples? This step may be able to increase the proportion of clonal mutations.
12. The development of APITH could be a good contribution. However, it is a simple metric by taking the averaging of heterogeneity for every pairwise combination of different regions of the

same tumor and it showed a high variation by the number of regions sequenced, which will be still impacted by the number of samples sequenced.

13. In the association to clinical outcomes, it is unclear how they defined low- and high-risk groups based on APITH? Did they use a median value or arbitrary value?

14. Does ITH associate with other clinicopathologic features like stage, tumor size, age etc.?

15. Was multivariate analysis done for the survival analysis adjusting for stage, tumor size, PS etc.? What if you used APITH to quantify ITH of TRACERx and correlate with their survival? Is their data comparable with yours?

16. The authors should be pay particular attention to the pathway analysis without gene expression data. Be sure to have reasonable cut offs for SCNA as well as methylation status as these can profoundly impact the genes selected.

17. Figure 4B, since the numbers of probes are not evenly distributed in all genome compartment and the majority of the probes in HM450K are designed to cover the promoter region and CpG island, so it is bias to compare ITH in different genomic compartments.

Added by Ed - paraphrased comments provided by Reviewer #2

1. The methodologies used as part of the study are not standard and that the study would have been strengthened using WES or WGS, and ideally with single cell seq.

2. The novel insights afforded by the study are minimal. Specifically that neither figure 1 nor 2 adds much information to the existing body of information on lung cancer genomics and evolution.

3. The benefits offered by APITH are presently unclear

4. The novelty of the data shown in figure 6 is unclear.

Reviewer #3 (Remarks to the Author):

In the manuscript 'Genetic and Epigenetic Intratumor Heterogeneity Impacts Prognosis of Lung Adenocarcinoma' by Hua X. et al. the authors calculated intratumoral heterogeneity (ITH) based on 292 tumor samples from 84 lung adenocarcinoma patients (LUAD) and 157 non-tumor samples (74 normal tissue samples, 81 blood samples, 2 buccal cell samples). All samples were treatment-naïve. Data was derived from deep targeted sequencing of 37 cancer driver genes, somatic copy number alterations (SCNA) from Illumina HumanOmniExpress SNP array and methylation from Illumina Infinium HumanMethylation450 BeadChip assays. Using these data sets for clonal architecture analyses they found strong evolutionary trajectories for SCNAs and DNA methylation. Furthermore, they developed an average pairwise ITH index (APITH) that is, in contrast to the calculations used in the TRACERx studies, independent of number of samples per tumor. Finally, they show that high ITH, regardless of SCNA or methylation, is associated with a poor prognosis. The study is important since the calculations of ITH become independent of sample sizes, which is a clear advantage over previous comparable studies. However, there are some major and minor concerns of the study:

Major concern:

- line 154 – 162 The authors describe a co-evolution of SCNA and DNA methylation. However, this could be due to a 'technical artefact' – e.g. by increasing the DNA amount in copy number gain regions, DNA methylation may purely be changed as a consequence of the changed ratio between alleles. This can not be resolved by Illumina Methylation arrays. How do the phylogenetic trees once DNA methylation has been adjusted for SCNAs? (see also PMID:29529299).

Minor concerns:

- Line 176 'Thus, we report below results on weighted analysis'. The conclusion is unclear.
- Why did the authors sequence only 37 genes and not whole exomes? This can give mutational and SCNA information at once.
- Figure 1: what do the numbers stand for? – labelling is unclear, where are tumor, where normal samples listed?
- Figure 2: why are there multiple lanes for each tumor?
- Figure 4: Where are the normal samples.

Reviewer #1 (Remarks to the Author):

In this interesting study, Hua et al investigated ITH of LUAD at SNV, SCNA and methylation level. Although the findings are similar to what have been reported by other groups, the development of the APITH approach is novel and could be a useful tool, particularly for studies on multiregion profiling of various number of samples per tumor. As such, this study is potentially interest to the community of heterogeneity of lung cancers. However, there are some technical and conceptual concerns.

1. Overall, the manuscript was well written and easy to follow. However, it may be too “concise” with inadequate information to assess the validity of the analyses. For example, it was not clear whether germ line DNA was used for SCNA analysis. It appears that SNP array was only applied in tumor sample. If so, it may not be appropriate to do SCNA segmentation using ASCAT without normal samples.

Response: We agree with the reviewer that in principal, there are several advantages in including matched normal tissue samples, e.g., to exclude germline CNV regions and obtain a better quantification of absolute copy numbers and LRR baseline. However, we profiled only tumor samples and took advantage of a specific feature in ASCAT that allows to perform segmentation without matched normal tissue samples (<https://www.crick.ac.uk/research/labs/peter-van-loo/software>). To address the issue of hyper-segmentation we modified the parameters in ASCAT to perform segmentation just based on BAF and then manually confirmed the segments. In addition, we only classified SCNAs into homozygous deletions, hemizygous deletions, LOH and amplifications based on LRR, without quantifying the absolute copy numbers for amplified regions, which would require matched normal tissues for best accuracy. To further verify the accuracy of our estimates, we now retrieved array-based genotyping data from blood (germline) samples for most patients included in this study who were previously included in genome-wide association studies (Landi et al., AJHG, 2009). We extracted LRR and BAF for these subjects and called CNVs using PennCNV (Wang et al., Nucleic Acids Research, 2008). Only very short CNVs were identified in the germline data and thus the reported SCNA events are not germline.

2. There are many interesting findings. However, the authors just described the data, but did not interpret the meaning. For example, the authors discovered “revealed that methylation ITH mapping to tumor suppressor genes was significantly lower than that of oncogenes and other genes (line 135-136)”. What is the interpretation of this finding?

Response: We have now added our interpretation of the data in the manuscript as follows:

“Inactivation of tumor-suppressor genes by hypermethylation at promoter regions has been observed in multiple cancer types including lung cancer¹⁷⁻¹⁹. Lower DNA methylation ITH in these regions suggests greater selective pressure which is consistent with their high putative impact in oncogenic transformation.”

3. In the unsupervised clustering with SCNA and DNA methylation profiles, how many patients showed the concordant clustering (all the regions of the same tumor cluster together)? It looks majority, but how many? This should be described in the text.

Response: We have included the precise numbers in the text. It now reads as follows:

For SCNAs: “Intratumoral heterogeneity was lower than intertumoral heterogeneity, with 226/268 (84.3%) samples from the same tumors clustered together and another 7/268 (2.6%) samples from the same tumors in close proximity to each other (Supplementary Fig. 2).”

For methylation: “Both analyses confirmed that normal tissue samples from almost all subjects (59/61) clustered together. Similarly, 183/205 (89.3%) samples from the same tumors clustered together, showing higher intertumoral heterogeneity than intratumoral heterogeneity.”

4. The figure legends for supplemental figures are too simple, which made it difficult to interpret the data.

Response: We thank the reviewer for this comment. We have now added more details in the figure legends of the supplemental figures.

5. The terms clonal and subclonal were not appropriate here. Since the authors did not do subclonal analysis to derive cancer cell fraction, they can not say whether the changes are clonal or subclonal. Based on the definition, trunk and branch are more appropriate terms.

Response: We agree with the reviewer that an alternative definition should be used in the absence of estimates of cancer cell fraction. We now used the terms ‘public’ to characterize genetic events carried by all tumor samples from the same tumors and ‘private’ for all other events. We did not use the terms ‘trunk’ and ‘branch’ to avoid confusion with the evolutionary trees described afterward.

6. It is not fair to compare the % of clonal vs subclonal SNVs with TRACERx study as the latter used PyClone to derive the clonality of SNVs. The authors can calculate the % trunk vs branch events from TRACERx publications for a fair comparison.

Response: We thank the reviewer for this suggestion. We now applied our criteria to quantify public and private SNVs in TRACERx data and reported the results in the text as follows: “65.3% (126 of 193) of SNVs were public, a higher proportion than what was observed in the TRACERx study⁶ (public SNVs = 50.5%), if we apply the same definition of public vs. private events.”

7. Ion Torrent is a good technology to validate known mutations. But it is not the best approach for “discovery” because of its much higher error rate. As such, one has to be very stringent to avoid technical artifacts. The filtering criteria “the variant allele count >3, coverage >2 in both tumor and normal samples and variant allele fraction ≥ 0.1 ” was relatively loose, which may lead to concerning proportion of SNVs are false positive mutations. There are some additional options. 1) Authors could use other software to call mutations and only include mutations called by more than one caller. 2) Authors should increase the sequencing depth and alternative read cut offs to increase the reliability. 3) Authors should pull out the TCGA data regarding these 37 genes to calculate the average number of non-synonymous mutations per sample to validate the findings are comparable.

Response: As the reviewer suggested, we re-examined our SNV calls. Most of the SNVs had high variant allele counts well beyond our filtering criteria. In total, we identified 500 SNVs (public SNVs in different samples were analyzed separately), and among them 98.4% had at least 10 variant alleles and 87.6% at least 50 variant alleles. This is likely due to the high sequencing depth (500x).

In addition, we analyzed non-synonymous mutations in the 37 genes in the TCGA LUAD patients and included the summary of the TCGA study in Supplementary Table S2. The mutation frequencies of the 37 genes in the two studies were highly correlated (Spearman’s $Rho=0.858$, $p<0.001$), although we found a higher frequency of variants likely because we had a much deeper coverage.

8. What is the definition of gain or loss? Is this based on Log Ratio? What cut off did you use, 0.3, 0.6, 1 or what?

Response: We added a detailed description of the methods we used to call SCNAs in the Methods section. It reads as follows:

“...we modified the ASCAT methods to rely only on BAFs for segmentation. Segments with BAF values different from 0.5 were defined as SCNA regions. We compared LRR values between SCNA regions and segments with BAFs=0.5 using the Student’s t-test. If the SCNA region LRRs were significantly higher or lower at significance level of 0.05 after adjusting for multiple comparisons, the segments were identified as amplified or deleted, respectively. Otherwise they were identified as LOH.”

9. The authors adjusted the methylation data by purity. What about SNCA data? Purity impacts SNCA profile too.

Response: We carefully adjusted the DNA methylation data by sample purity because the different levels of combination of normal and tumor samples can cause overestimation of ITH if unaddressed. For SCNA data, sample purity does not affect segmentation or the copy number status of large segments, which contribute the most to the ITH. While the absolute copy numbers could be affected by sample purity, the categorical status in large segments that we reported (SCNA or copy number neutral) is robust in samples with different purity.

In addition, we defined the copy number status by analyzing the absolute copy numbers which were estimated by ASCAT. In the statistical model of ASCAT, the Log R and BAF data are expressed as a function of the allele-specific copy number accounting for non-aberrant cell fraction (aka, sample purity) and tumor aneuploidy. Therefore, we adjusted for the impact of purity on the absolute copy number estimates using the ASCAT algorithm.

10. From deep-targeted sequencing, they found SNVs from 35 cancer driver genes. Are they all “cancer driver mutations”? Although they occurred in cancer driver genes, most of them can be passengers, not a driver mutation. It would be interesting if we can classify the mutations into “drivers” and “passengers” first and check what proportion of them is “clonal” and “subclonal”.

Response: We thank the reviewer for the suggestion. To distinguish driver vs. passenger mutations in cancer driver genes, we analyzed the predicted functional effects of the mutations and reported the results in the manuscript:

“In total, 24.3% (47/193; 30 public, 17 private) of SNVs were predicted to strongly alter protein functions (e.g., frameshift or gain of stop codon mutations). The public SNVs showed slightly higher dN/dS ratio than the private SNVs (public SNVs: 3.40 (95% CI: 1.80-6.44), private SNVs: 2.55 (95% CI: 1.29-5.05)). Overall, the functional impact and selective pressure showed no significant difference between public and private SNVs.”

11. The authors claimed that the proportion of clonal mutations was lower than in a previous TRACERx study. How did they deal with regions with low purity? Did they have any kind of rescue strategy for the mutations missing in the low purity samples? This step may be able to increase the proportion of clonal mutations.

Response: We conducted deep target sequencing of the cancer driver genes. Thus, even in the low purity samples, we should have been able to identify all SNVs. Thus, the definition of “clonal” (now “public”) mutations should be correct. Moreover, we now changed the criteria to define trunk and branches SNVs in the TRACERx data as suggested by the reviewer in #6. Based on our approach, our study shows now a larger number of public SNVs than TRACERx

12. The development of APITH could be a good contribution. However, it is a simple metric by taking the averaging of heterogeneity for every pairwise combination of different regions of the same tumor and it showed a high variation by the number of regions sequenced, which will be still impacted by the number of samples sequenced.

Response: We have made two contributions by introducing APITH.

- (A) The expectation of APITH does not depend on the number of tumor samples per patient; while the expectation of the naive ITH (NITH) metric does. When testing the association between NITH with a clinical outcome, spurious associations may be identified without appropriately controlling for potential hidden factors (e.g., tumor size or study sites) that are simultaneously associated with the clinical outcome and the number of tumor samples. Reducing the chance of spurious associations is one of the major concerns in population-based studies.
- (B) While the expectation of APITH estimates does not depend on the number of tumor samples, the variance of the estimates does, as was also noticed by the reviewer. While not impacting the type-I error rate when a predictor is measured with different levels of measurement error, it does impact the power of detecting associations. Thus, we introduced a weighted regression analysis with weights chosen to maximize the power to detect associations. Note that the optimal weight is dependent on the number of samples per subject. Subjects with more tumor samples are weighted more in regression analysis.

Thus, we believe APITH is appropriate for testing associations in a population-based study. We further maximized the power of detecting associations using subject-specific weights that depend on the number of tumor samples per subject.

13. In the association to clinical outcomes, it is unclear how they defined low- and high-risk groups based on APITH? Did they use a median value or arbitrary value?

Response: The low- and high- risk groups in the survival analysis were defined by the median APITH. We have now included the description in the Methods as follows: “KM-plot were stratified by the median APITH of subjects with at least two tumor samples.”

14. Does ITH associate with other clinicopathologic features like stage, tumor size, age etc.?

Response: We analyzed the association of ITH with other clinicopathological features and included the results in the manuscript as follows:

“We tested the association of SCNA and DNA methylation APITH with clinical data and observed no significant correlations with age, sex, stage or grade (Supplementary Table 4). Smokers had higher APITH of SCNAs ($P=0.035$, nominally significant) but similar APITH of methylation compared with non-smokers. Other smoking behaviors (e.g., smoking intensity and duration) were not associated with APITH of SCNA or DNA methylation.”

15. Was multivariate analysis done for the survival analysis adjusting for stage, tumor size, PS etc.? What if you used APITH to quantify ITH of TRACERx and correlate with their survival? Is their data comparable with yours?

Response: Our survival analysis was adjusted for stage, age and smoking status. We calculated APITH and used the same weighted Cox proportional hazard regression model to the SCNA data of the TRACERx study, adjusting for stage. It showed the same trend as in our study with a high SCNA APITH associated with poorer survival, but the association was not statistically significant in TRACERx (p -value=0.24, HR=1.35, 95% CI=0.82-2.20).

The TRACERx strong association ($p=0.0003$) between the naïve ITH index and survival that was reported in their manuscript Abstract was not adjusted for covariates. After adjusting for stage, the authors reported in the Results that the significance had dropped to 0.01.

16. The authors should be pay particular attention to the pathway analysis without gene expression data. Be sure to have reasonable cut offs for SCNA as well as methylation status as these can profoundly impact the genes selected.

Response: We thank the reviewer for this important comment. In response to the reviewer, and due to the unavailability of gene expression data, we adopted more stringent criteria to identify the potential SCNA and methylation driver events in the pathway analysis (Figure 5) and in the tumor evolutionary

trajectories (Figure 6). We have added a detailed description in the Methods Bioinformatics section as follows:

For SCNA Analysis: “In addition, amplifications with at least four copy numbers in oncogenes and deletions with zero copy number in tumor suppressor genes were identified as potential driver events.”

For DNA Methylation: “To identify potential driver DNA methylation events, we analyzed CpG island regions of cancer driver genes and compared the beta values of tumor samples and corresponding normal samples. We used 0.3 as the cutoff value to call differences in beta values⁷.”

17. Figure 4B, since the numbers of probes are not evenly distributed in all genome compartment and the majority of the probes in HM450K are designed to cover the promoter region and CpG island, so it is bias to compare ITH in different genomic compartments.

Response: We were aware of this possible bias and had scaled the distance by the number of CpG probes involved in ITH analysis. In fact, when we defined the distance for a pair of samples we used the following formula:

$$d_{ij} = \sqrt{\frac{\sum_{t=1}^T (x_{it} - x_{jt})^2}{T}},$$

We really appreciate the comment and we have now clarified this in the Methods.

Added by Ed - paraphrased comments provided by Reviewer #2

1. The methodologies used as part of the study are not standard and that the study would have been strengthened using WES or WGS, and ideally with single cell seq.

Response: We applied target sequencing primarily for two reasons. First, to provide a higher coverage depth in driver genes, which allows better detection of subclonal events. Second, given the lower cost, to assay more samples per subject, which provides a better landscape view of tumor ITH.

2. The novel insights afforded by the study are minimal. Specifically that neither figure 1 nor 2 adds much information to the existing body of information on lung cancer genomics and evolution.

Response: We disagree with the reviewer. Figure 1 is the study design. Our study involves analysis of multi-sampling and multi-platforms, this schematic overview is necessary to describe the number of samples for each assay used for this study.

Figure 2 described SNV clonality and provides an important illustration of lung cancer landscape.

3. The benefits offered by APITH are presently unclear

Response: As we summarized in our response to the first reviewer (12), our approach provided an important contribution. Specifically, we have made two key contributions by introducing APITH.

- (A) The expectation of APITH does not depend on the number of tumor samples per patient; while the expectation of the naive ITH (NITH) metric does. When testing the association between NITH with a clinical outcome, spurious associations may be identified without appropriately controlling for potential hidden factors (e.g., tumor size or study sites) that are simultaneously associated with the clinical outcome and the number of tumor samples. Reducing the chance of spurious associations is one of the major concerns in population-based studies.
- (B) While the expectation of APITH estimates does not depend on the number of tumor samples, the variance of the estimates does, as was also noticed by the reviewer. While not impacting the type-I error rate when a predictor is measured with different levels of measurement error, it does impact the power of detecting associations. Thus, we introduced a weighted regression analysis with weights chosen to maximize the power to detect associations. Note that the optimal weight is dependent on the number of samples per subject. Subjects with more tumor samples are weighted more in regression analysis.

Thus, we believe APITH is appropriate for testing associations in a population-based study. We further maximized the power of detecting associations using subject-specific weights that depend on the number of tumor samples per subject.

4. The novelty of the data shown in figure 6 is unclear.

Response: We disagree with the reviewer. Figure 6 showed the congruent evolutionary pattern of copy number and DNA methylation. This has never been shown in lung cancer before.

Reviewer #3 (Remarks to the Author):

In the manuscript 'Genetic and Epigenetic Intratumor Heterogeneity Impacts Prognosis of Lung Adenocarcinoma' by Hua X. et al. the authors calculated intratumoral heterogeneity (ITH) based on 292 tumor samples from 84 lung adenocarcinoma patients (LUAD) and 157 non-tumor samples (74 normal tissue samples, 81 blood samples, 2 buccal cell samples). All samples were treatment-naïve. Data was

derived from deep targeted sequencing of 37 cancer driver genes, somatic copy number alterations (SCNA) from Illumina HumanOmniExpress SNP array and methylation from Illumina Infinium HumanMethylation450 BeadChip assays. Using these data sets for clonal architecture analyses they found strong evolutionary trajectories for SCNAs and DNA methylation. Furthermore, they developed an average pairwise ITH index (APITH) that is, in contrast to the calculations used in the TRACERx studies, independent of number of samples per tumor. Finally, they show that high ITH, regardless of SCNA or methylation, is associated with a poor prognosis.

The study is important since the calculations of ITH become independent of sample sizes, which is a clear advantage over previous comparable studies. However, there are some major and minor concerns of the study:

Major concern:

- line 154 – 162 The authors describe a co-evolution of SCNA and DNA methylation. However, this could be due to a ‘technical artefact’ – e.g. by increasing the DNA amount in copy number gain regions, DNA methylation may purely be changed as a consequence of the changed ratio between alleles. This can not be resolved by Illumina Methylation arrays. How do the phylogenetic trees once DNA methylation has been adjusted for SCNAs? (see also PMID:29529299).

Response: We thank the reviewer for the great question and for pointing us to the important reference. We also agree that the Illumina methylation array does not have information to resolve this issue. To address this comment, we carried out a sensitivity analysis by restricting the analysis to CpG probes in copy number-homogeneous regions. Figure A here shows the scatter plot of SCNA APITH and methylation APITH using genome-wide CpG probes; Figure B shows the plot of SCNA APITH and methylation APITH using CpG probes in copy number-homogeneous regions; Figure C shows the plot of methylation APITH using genome-wide versus copy number-homogeneous CpG probes. The Spearman’s correlation coefficient was slightly increased from 0.59 to 0.64 (statistically not significant) by restricting CpG probes to regions with homogeneous copy number, still supporting the congruent evolution between SCNAs and methylation levels. Indeed, the methylation APITH in the two sets of probes are highly correlated. We have now included this sensitive analysis results in the manuscript as Supplementary Figure 5.

Minor concerns:

- Line 176 'Thus, we report below results on weighted analysis'. The conclusion is unclear.

Response: As we stated, we performed two analyses using APITH, one without weighting and the other weighted by the variance of APITH estimate that depends on the number of tumor samples per subject. We found that the weighted regression analyses provided more significant results, which is consistent with our theoretical investigation about the optimal weight to maximize the statistical power.

- Why did the authors sequence only 37 genes and not whole exomes? This can give mutational and SCNA information at once.

Response: We applied target sequencing primarily for two reasons. First, to provide higher depth in driver genes which allows better detection of subclonal events. Second, given the lower cost, we could assay more samples per subject, which provided a better landscape view of tumor ITH.

- Figure 1: what do the numbers stand for? – labelling is unclear, where are tumor, where normal samples listed?

Response: The numbers in figure 1 indicate the tumor samples from each subject profiled by different platforms. Figure 1 only describes the tumor samples. The detailed description of tumor and normal samples profiled by each platform is shown in Supplementary Figure 1. We now added more details in the figure legends.

- Figure 2: why are there multiple lanes for each tumor?

Response: The heatmap in figure 2 shows the SNVs in each sample, with each lane indicating one sample. Thick lines separate the different tumors. Multiple lanes within thick lines indicate multiple samples from the same tumor. We have now updated the figure legend as follows:

“Bottom panel: summary of intratumor heterogeneity for each gene and each tumor. Thick lines separate the different tumors. Multiple lanes within thick lines indicate multiple samples from the same tumor. “

- Figure 4: Where are the normal samples.

Response: The normal samples are indicated by the gray color in the column sidebar below the dendrogram. We have clarified this point within the legend as follows:

“...(A) Unsupervised hierarchical clustering of 5000 most variable probes in CpG islands of the genome in 68 subjects. Different tumors are indicated by different colors in the column sidebar, with normal samples colored in gray. The numbers in parenthesis are the number of normal tissue samples for the ‘normal’ group, or the number of tumor samples in each patient...”

Reviewers' comments:

Reviewer #2 (Remarks to the Author):

Unfortunately, I cannot recommend publication of this manuscript in Nature Communications. Overall, my sense is that the study lacks sufficient novelty for publication in a high-impact journal. Furthermore, I continue to believe that the technologies used in this study are not state-of-the-art anymore. In particular, the precise annotation of ITH in heterogenous cancers such as lung cancer requires definition of the CCF in a large number of mutations (e.g., by exome sequencing, or better, genome sequencing). I am not convinced that accurate ITH analyses can be performed by targeted sequencing. Changing the wording from "clonal" to "public" does not change this fact. Furthermore, copy number analyses are extremely limited when carried out through the use of arrays, primarily because of the physical signal saturation that is inherent to arrays, but also because of the limitations in resolution. This is why sequencing has essentially replaced array technology for determination of copy number states in cancer. Finally, I understand that the APITH approach has some value in population-based analyses; however, I feel strongly that this technology might be better suited for a subspecialty journal.

Reviewer #3 (Remarks to the Author):

The authors did not address my major concern of a just technical artefact of co-evolution of SCNA and DNA methylation. They need to reconstruct the evolutionary trajectories for methylation in SCNA-free areas. In addition, it remains unclear what they mean by 'homogenous copy number status'? Why is there still an increase in SCNA in Suppl. Fig.5B? If they use copy-number neutral regions then there should be no increase or decrease in CN. Otherwise this would still argue for a simple copy-number related increase in DNA methylation. As such this is no novelty, rather a methodological issue not solved so far. Minor points are addressed. The APITH is in general a useful tool for the study of ITH in varying numbers of samples.

Dear Editor,

We thank you for the opportunity to resubmit our response to reviewer 3 and address what we think was a mis-understanding. And we thank the reviewer for raising this question, which we think is very important and allowed us to improve our manuscript.

We copy below the exchange with the reviewer and our previous and new response, with related figures. We hope our response addresses the reviewer's concerns and the manuscript can be accepted for publication. We strongly believe this work offers an important contribution to the field.

Reviewer #3 (Remarks to the Author):

In the manuscript 'Genetic and Epigenetic Intratumor Heterogeneity Impacts Prognosis of Lung Adenocarcinoma' by Hua X. et al. the authors calculated intratumoral heterogeneity (ITH) based on 292 tumor samples from 84 lung adenocarcinoma patients (LUAD) and 157 non-tumor samples (74 normal tissue samples, 81 blood samples, 2 buccal cell samples). All samples were treatment-naïve. Data was derived from deep targeted sequencing of 37 cancer driver genes, somatic copy number alterations (SCNA) from Illumina HumanOmniExpress SNP array and methylation from Illumina Infinium HumanMethylation450 BeadChip assays. Using these data sets for clonal architecture analyses they found strong evolutionary trajectories for SCNAs and DNA methylation. Furthermore, they developed an average pairwise ITH index (APITH) that is, in contrast to the calculations used in the TRACERx studies, independent of number of samples per tumor. Finally, they show that high ITH, regardless of SCNA or methylation, is associated with a poor prognosis.

The study is important since the calculations of ITH become independent of sample sizes, which is a clear advantage over previous comparable studies. However, there are some major and minor concerns of the study:

Major concern:

- line 154 – 162 The authors describe a co-evolution of SCNA and DNA methylation. However, this could be due to a 'technical artefact' – e.g. by increasing the DNA amount in copy number gain regions, DNA methylation may purely be changed as a consequence of the changed ratio between alleles. This can not be resolved by Illumina Methylation arrays. How do the phylogenetic trees once DNA methylation has been adjusted for SCNAs? (see also PMID:29529299).

Response in 1st revision: We thank the reviewer for the great question and for pointing us to the important reference. We also agree that the Illumina methylation array does not have information to resolve this issue. To address this comment, we carried out a sensitivity analysis by restricting the analysis to CpG probes in copy number-homogeneous regions. Figure A here shows the scatter plot of

SCNA APITH and methylation APITH using genome-wide CpG probes; Figure B shows the plot of SCNA APITH and methylation APITH using CpG probes in copy number-homogeneous regions; Figure C shows the plot of methylation APITH using genome-wide versus copy number-homogeneous CpG probes. The Spearman's correlation coefficient was slightly increased from 0.59 to 0.64 (statistically not significant) by restricting CpG probes to regions with homogeneous copy number, still supporting the congruent evolution between SCNAs and methylation levels. Indeed, the methylation APITH in the two sets of probes are highly correlated. We have now included this sensitive analysis results in the manuscript as Supplementary Figure 5.

Reviewer's response to 1st revision:

Reviewer #3 (Remarks to the Author):

The authors did not address my major concern of a just technical artefact of co-evolution of SCNA and DNA methylation. They need to reconstruct the evolutionary trajectories for methylation in SCNA-free areas. In addition, it remains unclear what they mean by 'homogenous copy number status'? Why is there still an increase in SCNA in Suppl. Fig.5B? If they use copy-number neutral regions then there should be no increase or decrease in CN. Otherwise this would still argue for a simple copy-number related increase in DNA methylation. As such this is no novelty, rather a methodological issue not solved so far. Minor points are addressed.

The APITH is in general a useful tool for the study of ITH in varying numbers of samples.

Response: We appreciate that the reviewer raised this question again which gives us the opportunity to further clarify our analyses and conclusions. The reviewer rightly pointed out that the observed co-evolution or congruence between SCNAs and DNA methylation in many tumors could have been caused by SCNAs. While the Illumina methylation array does not have information to resolve this issue, a

sensitivity analysis restricted to CpG probes located in genomic regions that show constant SCNA profiles may rule out the possibility of a confounding effect.

In responding to the comment in the first-round review, we actually performed a sensitivity analysis using CpG probes in regions with the same copy number status (gain/loss/neutral) for all samples from the same tumor, defined as “SCNA homogeneous regions”. We also conducted the same analysis in copy-number neutral regions. Both analyses show very similar results, supporting the conclusion that the observed co-evolution between SCNA and DNA methylation was not confounded by the SCNA profiles. We observed that copy neutral regions only accounted for 22.5% of the genome for patients with ≥ 4 tumor samples, while SCNA homogenous regions accounted for 47.7% of the genome. Thus, we chose to report the results based on SCNA homogenous regions because they were more representative.

Here, we report the results based on both analyses. We also updated our manuscript reporting both sensitivity analyses and the related figures (**Supplementary Figure 5**).

In Figure 1 below, we report the scatter plots of pairwise Euclidean distance for each pair of tumor samples per tumor, calculated based on whole genome SCNAs and CpG probes in regions with the same copy number status (SCNAs homogenous regions) and copy-number neutral regions for all samples from the same tumor. Because each phylogenetic tree is built based on pairwise distances, high correlation of distances between two methods indicates similar phylogenetic trees of SCNAs and methylation. Based on these scatter plots, we observed that (1) DNA methylation distances calculated based on whole-genome CpG probes, copy number neutral regions and SCNA homogeneous regions are highly correlated with each other and (2) the correlations of pairwise distances between SCNA profiles and DNA methylation profiles are similar and strongly significant, no matter what CpG probes were used. The Figure 1B below is reported as **Figure 6A** in the manuscript (in comparison of the original version, we modified the description of the axes to be clearer). The other four panels (C to F) are in **Supplementary Figure 5 A-D**.

Figure I (related to **Figure 6A** and **Supplementary Figure 5** in the manuscript). Pairwise Euclidean distance of tumor samples from the same tumor based on (A) SCNAs and whole-genome CpG probes, (B) SCNAs and CpG probes in the SCNA homogeneous regions, (C) CpG probes in the SCNA homogeneous regions and whole-genome CpG probes, (D) SCNAs and CpG probes in copy number neutral regions, (E) CpG probes in copy number neutral regions and whole-genome CpG probes.

In the Figure II below, we report the same findings, but illustrated using phylogenetic trees based on CpG probes across the whole genome, CpG probes mapping to SCNA homogenous regions and CpG probes mapping to copy number neutral regions. The six tumors analyzed have at least five samples per tumor assayed for both SCNAs and DNA methylation, related to **Figure 6C** in the manuscript. Consistent with the high correlation of pairwise distances between different CpG categories in Figure 1, these trees are highly similar, further supporting that the SCNA status has minimal effect on the inference of phylogenetic tree analysis. We included these figures in **Supplementary Figure 5E**.

IGC-11-1044

Whole-genome methylation

CN-homoogeneous methylation

CN-neutral methylation

IGC-03-1007

Whole-genome methylation

CN-homoogeneous methylation

CN-neutral methylation

IGC-03-1101

Whole-genome methylation

CN-homoogeneous methylation

CN-neutral methylation

IGC-10-1178

Whole-genome methylation

CN-homogeneous methylation

CN-neutral methylation

IGC-10-1179

Whole-genome methylation

CN-homogeneous methylation

CN-neutral methylation

IGC-11-1090

Whole-genome methylation

CN-homogeneous methylation

CN-neutral methylation

Figure II. Phylogenetic analysis of tumors based on CpG probes across the whole genome (left), CpG probes mapping to SCNA homogenous regions (middle) and CpG probes mapping to copy number neutral regions (right). The six tumors have at least five samples per tumor assayed for both SCNAs and DNA methylation, related to **Figure 6C** in the manuscript.

REVIEWERS' COMMENTS:

Reviewer #3 (Remarks to the Author):

Thank you for the revised version of the manuscript. The authors have now addressed all my concerns.